# Therapeutic Potential of Bee and Wasp Venom in Anti-Arthritic Treatment: A Review

**DOI:** 10.3390/toxins16110452

**Published:** 2024-10-22

**Authors:** Hongmei Sun, Yunxia Qu, Xiaojing Lei, Qingzhu Xu, Siming Li, Zhengmei Shi, Huai Xiao, Chenggui Zhang, Zhibin Yang

**Affiliations:** 1Yunnan Provincial Key Laboratory of Entomological Biopharmaceutical R&D, College of Pharmacy, Dali University, Dali 671000, China; shm5620@163.com (H.S.); qyx1454@163.com (Y.Q.); l1458793853@163.com (X.L.); 15770266233@163.com (Q.X.); lsm7976@163.com (S.L.); szm18214194674@163.com (Z.S.); xiaohuai@dali.edu.cn (H.X.); 2National-Local Joint Engineering Research Center of Entomoceutics, Dali 671000, China

**Keywords:** bee venom, wasp venom, component, arthritis, mechanism

## Abstract

Arthritis has a high global prevalence. During the early ancient human era, bee (*Apis*) venom therapy was employed in Egypt, Greece, and China to alleviate ailments such as arthritis and neuralgia. In addition, bee venom has long been used as a traditional medicine for immune-related diseases in Korea. Wasp (*Vespa*) venom is a folk medicine of the Jingpo people in Yunnan, China, and has been widely used to treat rheumatoid arthritis. In spite of this, the underlying mechanisms of bee and wasp venoms for the treatment of arthritis are yet to be fully understood. In recent years, researchers have investigated the potential anti-arthritic properties of bee and wasp venoms. Studies have shown that both bee and wasp venom can improve swelling, pain, and inflammation caused by arthritis. The difference is that bee venom reduces arthritis damage to bone and cartilage by inhibiting the IRAK2/TAK1/NF-κB signaling pathway, NF-κB signaling pathway, and JAK/STAT signaling pathway, as well as decreasing osteoclastogenesis by inhibiting the RANKL/RANK signaling pathway. Wasp venom, on the other hand, regulates synovial cell apoptosis via the Bax/Bcl-2 signaling pathway, inhibits the JAK/STAT signaling pathway to reduce inflammation production, and also ameliorates joint inflammation by regulating redox balance and iron death in synovial cells. This review provides a detailed overview of the various types of arthritis and their current therapeutic approaches; additionally, it comprehensively analyzes the therapeutic properties of bee venom, wasp venom, or venom components used as anti-arthritic drugs and explores their mechanisms of action in anti-arthritic therapy.

## 1. Introduction

### 1.1. The Types of Arthritis

The global incidence rate of arthritis is substantial, encompassing more than 100 types, with rheumatoid arthritis (RA), osteoarthritis (OA), psoriatic arthritis, and inflammatory arthritis being the predominant classifications [1,2]. Arthritis is widely acknowledged to inflict significant harm to individuals. The progression of arthritis-induced pain is protracted and distressing [3]. Extensive research has demonstrated that the etiology of arthritis is multifactorial, including genetic predisposition, infectious agents, viral pathogens, and other contributing factors [4,5]. Despite an incomplete understanding of its pathogenesis, scientists have diligently pursued effective therapeutic strategies for patients afflicted by this condition [6].

Rheumatoid arthritis (RA) is a chronic inflammatory autoimmune disease characterized by joint swelling, increased joint pressure, and destruction of the synovial joint, which can lead to severe disability. The multifactorial etiology of rheumatoid arthritis primarily involves autoimmunity, infection, metabolic disorders, trauma, degenerative lesions, and other factors [7,8]. Approximately 0.5% of the population suffers from rheumatoid arthritis, with a female-to-male ratio of 2.5:1. The incidence rate increases with age, but most cases occur between the ages of 50 and 59 [8,9]. The treatment of rheumatoid arthritis has been revolutionized with the development of new classification criteria, the emergence of novel therapies, the introduction of early therapies, and the use of new effective strategies [8,10,11,12]. At present, disease-modifying anti-rheumatic drugs (DMARDs) serve as the primary treatment for RA. The goal in treating arthritis is not only achieving disease remission but also improving the functional state [12,13]. Despite extensive research on its pathogenesis and pathological characteristics [14,15], there is currently no perfect preventive measure or permanent cure for RA after disease onset.

Osteoarthritis (OA) is a chronic osteoarthropathy that arises from various factors, including strain, obesity, trauma, excessive exercise, congenital joint abnormalities, deformities, and heredity; it is primarily characterized by degenerative changes in articular cartilage, accompanied by subchondral osteosclerosis and hyperplasia. Gradual progression of joint pain, swelling, stiffness, restricted mobility, and deformity are distinguishing features of OA. This condition has consistently affected a significant portion of the population, and its prevalence is closely associated with aging and obesity rates. The population affected by osteoarthritis is predominantly elderly, with the majority over the age of 65, and a higher proportion of women than men suffering from the disease [16]. Current pharmacological treatments for OA (e.g., oral nonsteroidal anti-inflammatory drugs, intra-articular corticosteroid injections, and lubricants) only provide symptomatic relief; they do not stop the progression of OA [17,18].

Psoriatic arthritis (PsA) is a chronic and progressive from of arthritis characterized by inflammation related to psoriasis, primarily affecting the spine or peripheral joints [19]. Reports indicate that psoriatic arthritis typically affects 0.1–1% of the general population and about 20% of people with psoriasis [20]. Approximately 30% of people with psoriasis are diagnosed with psoriatic arthritis [21,22]. The incidence of PsA increases progressively with the duration of the disease, and clinical symptoms are associated with arthritis, including swelling, stiffness, and limited mobility, as well as underlying manifestations of sacroiliac arthritis, spondylitis, dactylitis, and attachment point inflammation, where joint destruction can lead to deformity and disability within two years of diagnosis [23,24]. The most effective drugs for the treatment of psoriatic arthritis include nonsteroidal anti-inflammatory drugs (NSAIDs), corticosteroids, and interleukin-17/23 (IL-17/23), interleukin-17 (IL-17), and CTLA-4 inhibitors. However, these drugs are deficient in the treatment of psoriatic arthritis or may bring about side effects; for example, NSAIDs do not ameliorate the skin damage or prevent the joint damage caused by PsA or stop the progression of the arthritis, glucocorticoids can induce a variety of complications, and the rest of the medications may cause adverse effects, such as local injection site reactions, gastrointestinal upset, and upper respiratory tract infections [25,26]. Due to the high prevalence of PsA among psoriasis patients, new therapeutic agents are urgently needed to manage this condition [27].

Gout is characterized by the accumulation of monosodium urate (MSU) in the synovium or periarticular tissues, representing a prevalent arthritic condition [28]. Severe cases of gout typically manifest with symptoms such as excruciating pain, erythema, joint redness, and swelling [29,30]. Failure to promptly treat gout can result in joint stiffness and deformity, leading to increased financial burden and significant impairment of quality of life [31,32]. Considering the pathogenesis of gout, the timely inhibition of inflammation and reduction in serum uric acid levels are currently considered to be the most effective treatment strategies [30,33,34]. For this reason, alternative treatments for hyperuricemia and gouty arthritis are essential.

Juvenile idiopathic arthritis (JIA), a chronic rheumatic disease of childhood that commonly affects children under the age of 16, has an unknown etiology and currently lacks a cure [35,36,37]. The treatment strategy for JIA involves the progressive use of NSAIDs or intra-articular corticosteroid injections, with the goal of achieving disease remission [38]. There are a growing number of therapeutic options for JIA, including IL-1, IL-6, JAK inhibitors, and biologics that simultaneously target IL-18, interferon gamma, or IL-1β and IL-18. However, the treatment of JIA remains a challenging dilemma for rheumatologists [39]. While appropriate treatment can lead to significant improvement in patients’ condition, inappropriate or untimely treatment may result in joint damage or irreversible disability [40,41], which can be extremely distressing for pediatric patients and have negative impacts on their psychological and physical health and development. Therefore, developing new therapeutic drugs is essential to improve outcomes for individuals with JIA [42]. 

Ankylosing spondylitis (AS), characterized by inflammation and structural damage to the sacral joints or spine, mainly affects young people, is a chronic inflammatory disease that affects 0.1–0.5% of the global population, and is a persistent rheumatic disease [43,44]. Typically, this inflammatory process causes patients to experience persistent back pain and morning stiffness which, in severe cases, can lead to limited spinal mobility [45,46]. The distinguishing feature of ankylosing spondylitis lies in the development of bone spurs, setting it apart from other rheumatic diseases. Consequently, the primary objective of treating ankylosing spondylitis revolves around controlling new bone formation. However, limited data exist regarding the efficacy of treatment for compulsive arthritis. Existing treatment options may reduce symptoms but are not effective in stopping or reversing physical damage [47,48].

The most common types of anti-arthritis medications are anti-rheumatics, nonsteroidal anti-inflammatory drugs, glucocorticoids, and biologics. However, the relatively high cost and serious side effects of these therapies make them poor treatment prospects for arthritis patients, and new therapies are urgently needed [49,50]. Traditional Chinese medicine has been shown to be effective in the treatment of arthritis, relieving joint pain and dysfunction and improving overall quality of life [51]. A variety of insect-derived medicines have been reported for the treatment of arthritis, including honeybees, which have been widely used in the treatment of chronic pain disorders since ancient times, and wasp venom, which is used as a traditional medicinal herb by the Jingpo ethnic group of Yunnan Province for the treatment and prevention of rheumatoid disorders [52,53,54,55,56].

### 1.2. Bee (Apis) Venom

Bee venom is a colorless and bitter liquid containing toxic peptides, proteins with an enzymatic role, and amines secreted from the poison glands of bees [57]. Undoubtedly, bee venom (BV) plays a crucial role in safeguarding bee colonies from predators and stands as a unique weapon in the animal kingdom; it is an intricate and highly effective blend of ingredients designed to protect bees against various predators, ranging from arthropods to vertebrates [58]. Bee venom’s medicinal history dates back to ancient Egypt and ancient Greece, while its usage dates back 3000–5000 years in China. Bee venom has been used in Traditional Korean medicine for centuries [55]. Bee venom has also demonstrated promising results as a biological therapy in several clinical experiments. On the one hand, it exhibits high efficacy in combating inflammation and destroying connective tissue; on the other hand, it aids in restoring mobility and flexibility by supporting the body’s natural defense mechanisms [59]. Bee venom can enter the human body through direct stings or manual injection [60]. BV has been used in traditional medicine to treat chronic pain diseases and alleviate discomfort [56]. Researchers have shown that both humans and animals can benefit from BV therapy for its therapeutic potential [61]. According to some studies, bee venom exerts both pharmacological and mechanical effects through the interaction of bioactive compounds in the venom and acupuncture stimulation [62]. Bioactive compounds in bee venom include melittin (the main active ingredient), apamin, adolapin, mast cell degranulation peptide, phospholipase A2, and so on. Nevertheless, their safety is still a key issue in practical application, and their mechanisms of action and pharmacological characterization need to be further investigated [63,64,65]. 

### 1.3. Wasp (Vespa) Venom

The common wasp species include *Vespa mandarinia* [66], *Vespa affine* [67], *Vespa ducalis* [67], and *Vespa magnifica* [68]. Notably, the crude venom of *Vespa mandarinia* has demonstrated a cardioactive effect on rat hearts in electrocardiogram experiments [66,67], while the aqueous extract of *Vespa affinis* has exhibited antioxidant capabilities [69]. Additionally, the venom of both *Vespa ducalis* and *Vespa affinis* has demonstrated potent antimicrobial effects [67]. Among the various wasp species, the therapeutic potential of *Vespa magnifica* in treating arthritis has garnered significant attention.

*Vespa magnifica* (Smith) is native to Yunnan, China, and is widely used in the Dehong Jingpo region of Yunnan for the treatment and prevention of rheumatic diseases, among other things. This is a local ethnic remedy, and its liquor is included in the *Chinese Pharmacopoeia* [69]. To date, our group has isolated and identified four active components of wasp venom—namely, 5-hydroxytryptamine, vespakinin-M (VK), mastoparan-M, and vespid chemotactic peptide-M—and has carried out activity studies on them [68]. The treatment of rheumatoid arthritis with wasp venom is currently rare, but several studies have found that this treatment has a beneficial effect [68,70,71,72,73].

In the present review, we synthesize the relevant studies on bee venom and wasp venom for the treatment of arthritis, and we describe the mechanisms of action associated with these venoms and their main components (Figure 1).

## 2. Anti-Arthritic Effects of Bee Venom and Its Major Components

### 2.1. Anti-Arthritic Effects of Bee Venom

Evidence shows that bee venom plays an important role in the inflammatory response by reducing cytokine and chemokine expression, alleviating synovitis and neutrophil infiltration, and significantly ameliorating ankle swelling and abnormal pain in gouty rats [75]. In a study by İbrahim Tekeoğlu et al. (2020), significant improvements in arthritis-related symptoms and inflammatory markers, including IL-1β, TNF-α, and IL-6, as well as increases in total oxidant levels (TOL) and oxidative stress index (OSI), were observed in the group administered bee venom at doses of 2 µg/kg, 4 µg/kg, and 20.0 μg/kg. Bee venom treatment did not cause significant changes in hepatotoxicity or liver function indices (ALT and AST), and it effectively reduced the symptoms of adjuvant arthritis in experimental rats [76]. In addition, a study by Eun Ju Im et al. (2016) showed that bee venom pretreatment of LPS-induced BV2 microglia significantly reduced the rate of NO production, decreased the mRNA expression levels of COX-2, TNF-α, IL-1β, and IL-6, and effectively inhibited LPS-induced activation of MyD88 and IRAK1 as well as phosphorylation of TAK1. These findings suggest that the anti-inflammatory effects of bee venom may be mediated through the IRAK1/TAK1/NF-κB signaling pathway (Table 1) [77].

### 2.2. Anti-Arthritic Effects of the Main Components of Bee Venom

Bee venom has been utilized in Eastern medicine for over 3000 years as a natural anti-inflammatory agent. In total, 88% of BV is water, and most of the rest is proteins and peptides, with about 162 proteins and peptides (Appendix A), including melittin, apamin, adolapin, and MCD peptides, in addition to enzymes, biologically active amines, and nonpeptide components (Table 2) [57].

According to Woon-Hae Kim et al., bee venom primarily consists of melittin, a linear peptide comprising 26 amino acid residues that constitutes 50% of the venom’s composition [81]. Various studies have found that the anti-inflammatory properties of bee venom and its main component melittin can be attributed to their interaction with the sulfhydryl group of NF-κB. The results showed that the binding of melittin to the P50 and p65 sulfhydryl groups of NF-κB resulted in IKK activation, IκB release, reduction in NF-κB activity, and subsequent production of inflammatory mediators. In addition, since the P50 sulfhydryl group is located on IKK, IKK can be considered as a potential target for the anti-inflammatory effects of BV, while melittin can exert pro-apoptotic effects by inhibiting the activation of STAT transcription factors and regulating mitochondrial apoptosis-related genes to explore the possible pro-apoptotic effects of bee venom peptides on IL-6/sIL6R-stimulated human fibroblast-like synoviocytes (FLS). Inflammatory stimuli cause fibroblast-like synoviocytes to proliferate abnormally and release inflammatory mediators, thereby exacerbating the inflammatory response and making fibroblast-like synoviocytes an important target for the treatment of arthritis [82,83,84]. According to Park et al. (2007), both BV and melittin were found to block LPS-induced expression of dCOX-2, cPLA2, and iNOS. These findings suggest that the anti-inflammatory properties of BV and, melittin play an important role in the effectiveness of the treatment of arthritis [82]. A study conducted by Choe and Kim (2017) examined the effects of bee venom on bone metabolism and osteoclast formation. Osteoclasts can lead to osteoporosis and joint destruction. In patients with arthritis, osteoclast activity is increased, exacerbating disease progression and worsening symptoms. Therefore, inhibiting osteoclast activity has become one of the most important strategies for the treatment of arthritis. The results of the study showed that bee venom hindered the generation of osteoclast-like monocytes by disrupting the RANKL-RANK signaling pathway [85]. Therefore, since, melittin can inhibit the production of inflammatory mediators, induce synoviocyte apoptosis, and inhibit osteoclastogenesis, it is concluded that it has a promising future for the treatment of RA.

Apamin (APM), a secondary component of bee venom, constitutes approximately 2–3% of its contents. This polypeptide consists of 18 amino acid residues [86]. Several studies have demonstrated that apamin exhibits anti-inflammatory properties in a variety of animal models of inflammatory diseases, and that its anti-inflammatory mechanism involves the inhibition of cyclooxygenase-2 and phospholipase A. In arthritic mice, APM plays a key role not only by inhibiting the production of inflammatory cytokines, such as TNF-α, IL-1β, and IL-6, but also through attenuating the paw swelling and MSU-induced pain in gouty mice (Table 2 and Figure 2) [87].

Phospholipase A2 (Bvpla2), which makes up about 10 to 12% of bee venom, plays a key role in the inflammatory process. Bvpla2 plays a role in the inflammatory process through modulating arachidonic acid release and arachidonic acid-like production, and it may also reduce collagen-induced polyarthritis in mice by promoting the polarization of Foxp3+ regulatory T cells (Table 3 and Figure 2) [88,89].

## 3. Anti-Arthritic Effects of Wasp Venom and Its Major Components

### 3.1. Anti-Arthritic Effects of Wasp Venom

Wasp venom possesses anti-inflammatory, analgesic, and immunomodulatory properties [90]. In a previous study by our group, it was found that wasp venom extracts exhibited excellent anti-arthritic effects in CIA rats, which can be attributed to their immunomodulatory and anti-inflammatory properties [91].

Additional experiments have demonstrated that the venom of wasps has the ability to manage synovial inflammation and provide treatment for RA through obstructing activation of the JAK/STAT signaling pathway [70]; moreover, it downregulates the expression of inflammatory factors induced by tumor necrosis factor-α (such as IL-1β and IL-6) and enhances apoptosis via the Bax/Bcl-2 signaling pathway within mitochondria [70,72,91]. In a recent study conducted by Ni et al. in 2023, it was shown that WV (wasp venom) and WV-II (3–10 kDa wasp venom) had inhibitory effects on the inflammatory response and cell proliferation of MH7A cells. These venoms modulated the JAK/STAT signaling pathway, redox homeostasis, and ferroptosis. Unlike apoptosis, necrosis, and autophagy, ferroptosis is a unique type of controlled cell death that depends on iron and reactive oxygen species (ROS). The impact of ferroptosis inducers on glutathione peroxidases (GPXs) leads to diminished cellular antioxidant capability, accumulation of ROS and, ultimately, oxidative cell death (Table 4 and Figure 3) [70,92]. These investigations provide an experimental basis for the future advancement of bioactive monomers derived from wasp venom.

### 3.2. Potential Anti-Arthritic Effects of the Main Components of Wasp Venom

It has been found that wasp venom contains a number of active components, usually proteins and peptides, with approximately 96 protein and peptide components (Appendix A). To date, researchers have successfully isolated and identified four active ingredients, specifically: 5-hydroxytryptamine, vespakinin-M (VK), mastoparan-M, and vespid chemotactic peptide-m (Table 5) [67,68].

Vespakinin-M (VK) is a hydroxyproline-containing bradykinin (BK) analog, initially reported in the 1970s after being isolated from the venom of Vespa mandarinia. The findings suggest that VK promotes functional recovery in mice after ischemic stroke, including amelioration of neurological damage, reduction in infarct volume, maintenance of the blood–brain barrier integrity, and blockade of inflammatory responses and oxidative stress. Furthermore, VK treatment resulted in reduced neuroinflammation and apoptosis associated with activation of PI3K/AKT and inhibition of the IκBα-NF-κB signaling pathway (Table 6) [90]. The anti-inflammatory properties exhibited by VK endow it with the potential to treat various types of arthritis. This potential anti-arthritic effect provides a useful direction for future research and development. Thus, VK may become an important candidate for future arthritis therapeutic strategies of great interest, providing strong support for exploring new therapeutic avenues and drug development.

Mastoparan-M (Mast-M), constituting approximately 70–80% of the crude venom, represents the predominant component of wasp venom. This amphipathic peptide toxin falls under the category of cell-penetrating peptides [73,93]. It has been shown that Mast-M significantly ameliorates gouty arthritis, and the underlying mechanism may be the inhibition of NLRP3 inflammasome activation by inhibiting the MAPK/NF-κB pathway and attenuating oxidative stress (Table 6) [56]. Mast-M has demonstrated good results in improving gouty arthritis, providing a good entry point for the future development and utilization of Mast-M in the treatment of other types of arthritis, as well as establishing a solid foundation.

Vespid chemotactic peptide-M has rarely been reported, but it exhibits potent antimicrobial activity against both bacteria and fungi [94]. Future studies on Vespid chemotactic peptide-M may gradually increase, and it is expected to find that it exhibits more significant anti-inflammatory or anti-arthritic effects, and it is expected that future studies will reveal more information about its potential role.

## 4. Conclusions and Future Prospects

Bee (*Apis*) and wasp (*Vespa*) venoms have received much attention in previous studies as potential natural substances for the treatment of arthritis. These venoms have been found to have multiple effects in inflammation and immunoregulation, including interfering with the RANKL-RANK signaling pathway, inhibiting the activation of transcription factors (especially STAT3 and NF-κB p65), regulating the expression of genes associated with mitochondrial apoptosis, regulating arachidonic acid metabolism and inducing polarization of regulatory T cells, inhibiting the JAK/STAT signaling pathway, decreasing inflammatory factor (e.g., IL-1β, IL-6, TNF-α) levels, and so on. These combined effects provide a multifaceted therapeutic strategy and research direction for the treatment of arthritis, which is expected to alleviate the pain and discomfort caused by arthritis and provide a scientifically documented therapeutic option for patients with arthritis, especially for those who have a limited response to traditional treatments.

Further studies could provide insight into the mechanism of action of bee venom and wasp venom in arthritis treatment. Previous studies have found that high concentrations of bee venom are potentially toxic to normal non-target cells and tissues, e.g., affecting cellular DNA stability, disrupting cell membranes, or producing hemolysis, and these studies are critical to validate the efficacy and safety of these venoms in arthritis treatment [95]. Drug distribution during administration is difficult to differentiate, compromising therapeutic efficacy and increasing the risk of side effects. An ideal drug delivery system should overcome these obstacles and maximize the effect of the encapsulated drug. Studies have shown that nanocarriers can increase drug solubility, prolong circulation time, reduce clearance, and deliver precisely to the disease site. This suggests that we can improve the precision and effectiveness of venom therapy by combining these venoms with new technologies such as nanotechnology tong technology, which is expected to improve the precision and effectiveness of venom therapy while reducing potential side effects, resulting in a better treatment experience for arthritis patients [96]. The development of a bee venom nanoemulsion has been shown to have an anti-inflammatory effect on a rat model of type II collagen-induced arthritis, effectively reducing inflammation in the foot and paw of arthritic rats [97]. However, animal models are limited and cannot fully mimic human disease. Although nanocarriers are versatile, they are complex to synthesize, biocompatible, difficult to predict in vivo behavior, and difficult to produce on a large scale, so hornet venom is still a long way from clinical application.

In addition, as we continue to deepen our research on bee and hornet venom, we should also focus on their long-term effects and safety in arthritis treatment. At the same time, interdisciplinary collaboration and technological innovation can be leveraged, and we hope to develop more personalized treatment protocols to provide comprehensive support for the treatment of arthritis patients.

Overall, bee venom and wasp venom have a promising future as potential therapeutic agents for the treatment of arthritis. Through continued research endeavors, we expect to further reveal the therapeutic mechanisms of these natural substances and introduce more innovative and effective treatment options for arthritis patients. In our future explorations, we look forward to seeing these venoms play a greater role in clinical practice and contribute to further improving the quality of life of arthritis patients.

## 5. Methodology

A literature search of original articles was conducted to identify research review articles related to many types of arthritis, bee venom versus wasp venom, and their main components for the treatment of arthritis. Multiple online databases were searched, including Google Scholar, PubMed, CNKI, and Web of Science. As of 2024, we have retrieved a total of 445 relevant Chinese and English pieces of literature on the anti-arthritis effects of bee venom and wasp venom. After excluding 114 duplicate articles, we finally identified 313 articles related to the anti-arthritis effects of bee venom. There were 18 articles related to the anti-arthritis effects of wasp venom. Accessed on 1 August 2024 the Uniprot database (https://www.uniprot.org/) with the search strategy (https://www.uniprot.org/uniprotkb?query=Apis+AND+venom) and (https://www.uniprot.org/uniprotkb?query=Vespa+AND+venom), we conclude that bee venom contains 162 protein and peptide components, with the remaining type of about 30, of which about three components are in high concentrations and can be used as anti-arthritic drugs; wasp *(Vespa)* venom contains 96 protein and peptide components, and after purification, four components were isolated, three of which have potential anti-arthritic effects.

The following keywords were queried individually and in combination to identify studies relevant to this review: “bee venom”, “wasp venom”, “arthritis”, “rheumatoid arthritis”, “osteoarthritis”, “psoriatic arthritis”, “juvenile idiopathic arthritis”, “Ankylosing Spondylitis”, “Traditional Applications”, “Melittin”, “Apamin”, “Vespakinin-M (VK)”, “Mateparae-M”, “Vespid chemotactic peptide-M”, “mechanism of action”, and others as needed. We included articles and books relevant to the topic of this review, ultimately documenting nearly 100 relevant references.

The main points of this review of the main components of bee venom and wasp venom and their mechanisms of action in the treatment of arthritis are listed in the charts and text. Accessed on 14 August 2024 the House of Researchers website (https://www.home-for-researchers.com/#/) to draw all figures in the text online.

## Figures and Tables

**Figure 1 toxins-16-00452-f001:**
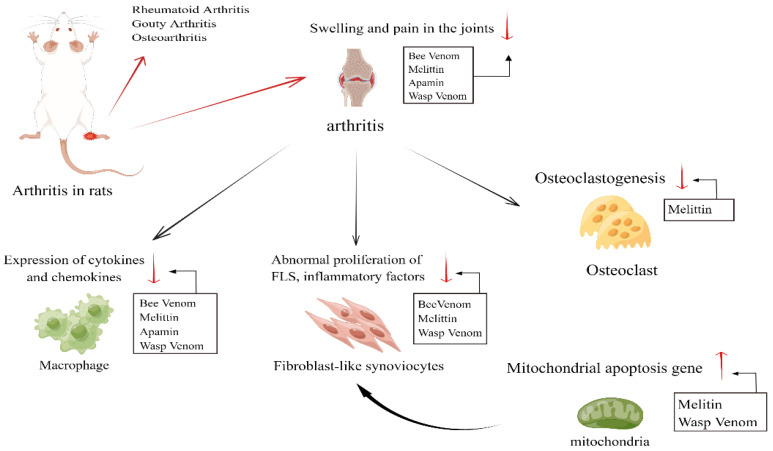
Primary therapeutic targets of wasp and bee venoms in the context of arthritis treatment [70,74].

**Figure 2 toxins-16-00452-f002:**
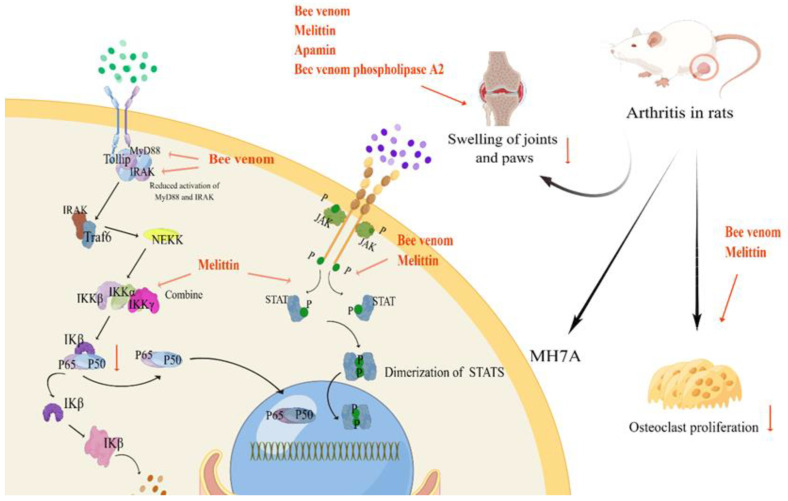
Primary targets of bee venom for the treatment of arthritis [75,82,83,85,87,88,89].

**Figure 3 toxins-16-00452-f003:**
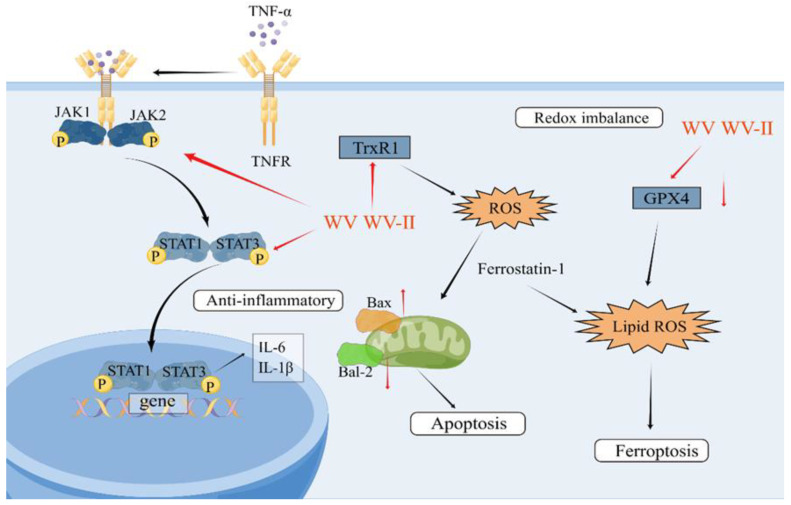
Main targets of wasp venom for the treatment of arthritis [70,92].

**Table 1 toxins-16-00452-t001:** Mechanisms of anti-arthritic action of bee venom.

Component	Mechanisms
Bee venom	1. Reduces cytokine and chemokine expression and attenuates synovitis and neutrophil infiltration [75].
	2. Downregulates inflammatory markers IL-1β, TNF-α, and IL-6, and upregulates total oxidant level (TOL) and oxidative stress index (OSI) [76].
	3. Inhibits IRAK1/TAK1/NF-κB signaling pathway [77].

**Table 2 toxins-16-00452-t002:** Components of bee venom and their major characteristics [57,78,79,80].

	Component	MW	Percent in Dry Venom (%)	Major Characteristics	Source (Uniprot ID)	Organism
Peptides	Melittin	2840	40–50	Containing 26 amino acid residues, inhibiting tumor growth, reducing inflammation, and alleviating arthritis symptoms.	P0DPR9; P01501; Q8LW54; P01502; P01504; P68407	*Apis cerana*; *Apis mellifera*; *Apis dorsata*; *Apis florea*;
	Apamin	2036	1–3	Containing 18 amino acid residues, reducing inflammation, and cytotoxic effect against cancer.	P01500; Q86QT2	*Apis mellifera*; *Apis cerana*
	MCD (mast cell degranulation)	2588	1–2	Containing 22 amino acid residues, relief of pain, and reducing inflammation.	P01499; P04567; Q6H2Z4	*Apis mellifera*; *Bombus pensylvanicus (American bumblebee) (Apis pensylvanica)*; *Apis cerana*
	Secapin	2755	0.5–1	Containing 24 amino acid residues, induction of leukotriene-mediated nociceptive hypersensitivity and edema.	Q7YWB0; C0HLU0; I1VC85; A0A0K1YW63; P02852	*Apis cerana*; *Apis mellifer;*
	Pamine		1–3			*Apis*
	Minimine	6000	2–3			*Apis*
	Procamine A and B	600	1–2	Containing 5 amino acid residues.		*Apis*
	Protease inhibitor	9000	<0.8			*Apis*
	Cardiopep	2500	<0.7			*Apis*
	Melittin F	2239	<1	Containing 19 amino acid residues.	P0DPR9; P01501; Q8LW54; P01502; P01504; P68407	*Apis cerana*; *Apis mellifera*; *Apis dorsata*; *Apis florea*
	Adolapin		1	Containing 103 amino acid residues, anti-nociceptive effect, reducing inflammation, and antipyretic effects.		*Apis*
	Tertiapin	2497	<0.1	Containing 21 amino acid residues, high-affinity inhibitor of inwardly rectifying potassium channels.	P56587	*Apis mellifera*
Enzymes	Phospholipase A2	15,000–16,000	12–15	Containing 128 amino acid residues, inhibiting tumor growth and proliferation, and reducing inflammation.		*Apis*
	Hyaluronidase	38,000	1–3			*Apis*
	Acid phosphomonoesterase	55,000	1			*Apis*
	α-Glucosidase	170,000	0.6			*Apis*
	Lysophospholipase		1			*Apis*
Phospholipids		700	1–3			*Apis*
Amines	Histamine	307.14	0.5–2			*Apis*
	Dopamine	189.64	0.13–1			*Apis*
	Noradrenalin	169.18	0.1–0.7			*Apis*
	Neurotransmitters		0.1–1			*Apis*
Amino	γ-Aminobutyric acid	189.64	0.13–1			*Apis*
	α-Amino acids	169.18	0.1–0.7			*Apis*
Carbohydrates	Glucose	180	2–4			*Apis*
	Fructose					*Apis*
Minerals	Phosphate		3–4			*Apis*
	Calcium					*Apis*
	Magnesium					*Apis*

**Table 3 toxins-16-00452-t003:** Mechanisms of action underlying primary components of bee venom.

Component	Mechanisms
Melittin	1. The sulfhydryl group of p50 in NF-κB is engaged in a reaction [82].
	2. Melittin binding to the sulfhydryl group of IKKs [75].
	3. The effects of bee melittin on iNOS, COX-2, and cPLA2 expression [82].
	4. Interfering with the RANKL-RANK signaling pathway [85].
	5. The suppression of transcription factor activation, specifically STAT3 and NF-κB p65, along with the modulation of genes associated with mitochondrial apoptosis [83].
Apamin	1. Inhibition of cyclooxygenase-2 and phospholipase A2 [87].
	2. Repression of the production of pro-inflammatory cytokines TNF-α, IL-1β, and IL-6 [87].
Phospholipase A2	1. Modulating the release of arachidonic acid and the generation of eicosanoids [88].
	2. Inducing the polarization of murine Foxp3+ regulatory T cells [89].

**Table 4 toxins-16-00452-t004:** Mechanisms of anti-arthritic action of wasp venom.

Component	Mechanisms
Wasp venom	1. Inhibition of JAK/STAT signaling pathway activation [70].
	2. Downregulation of the expression of inflammatory factors (e.g., IL-1β, IL-6) [72].
	3. Reinforcing apoptosis via the mitochondrial Bax/Bcl-2 signaling pathway [91].
	4. Rebalancing redox and triggering ferroptosis in synovial fibroblasts [70,92].

**Table 5 toxins-16-00452-t005:** Components of wasp venom and their major characteristics.

	Component	MW	Major Characteristics	Source (Uniprot ID)	Organism
Biogenic amines	5-Hydroxytryptamine	177.0768	5-Hydroxytryptamine (5-HT) is an inhibitory neurotransmitter that regulates various physiological processes in the central nervous system and peripheral tissues.		*Vespa*
Peptides	Vespakinin-M	1361.6845	Containing 12 amino acid residues, a novel bradykinin analog containing hydroxyproline.	Q0PQX8; Q7M3T3	*Vespa magnifica*; *Vespa mandarinia*
	Mastoparan-M	1478.9717	Containing 14 amino acid residues, accounting for 70–80% of crude venom.	P04205	*Vespa mandarinia*; *Vespa magnifica*
	Vespid chemotactic peptide-M	1382.8684	Containing 13 amino acid residues.	P17232	*Vespa mandarinia*; *Vespa magnifica*

**Table 6 toxins-16-00452-t006:** Potential anti-arthritic mechanism of action of wasp venom components.

Component	Mechanisms
Vespakinin-M (VK)	1. Activation of PI3KAKT and inhibition of the IκBα-NF-κB signaling pathway [90].
Mastoparan-M	1. Inhibition of MAPK/NF-κB pathway and attenuation of oxidative stress to suppress NLRP3 inflammasome activation [56].

## Data Availability

No data were utilized in the research conducted as described in the article.

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
