# Peer review of "Therapeutic Potential of Bee and Wasp Venom in Anti-Arthritic Treatment: A Review"

_toxins, 2024, doi:10.3390/toxins16110452_

Round 1

Reviewer 1 Report (New Reviewer)

Comments and Suggestions for Authors

The manuscript titled “Therapeutic Potential of Bee and Wasp Venom in Anti-Arthritic Treatment: A Review” aimed to summarized the significant potential of bee and wasp venoms in the treatment of arthritis.

In the Abstract section please briefly summarise the results from your literature finding. How do these two venoms actually impact arthritis and by which mechanism, based on the current literature? In its current form, the Abstract is rather vague.

In the Introduction section please give more information on the composition of both bee and wasp venom. The author could list either in the text or in the Table/Chart the main constituents of bot venoms such as their peptides, enzymes and so on. Now I see that such tables are incorporated later in the paper but I would advise to follow the same outline for both bee and wasp venom. E.g. first put the table regarding components of the venom itself and then the ones regarding the mechanisms of anti-arthritic action.

Authors could discuss the possible toxicity of either bee or wasp venom to normal non-target cells if used for the proposed treatment. Although numerous animal venoms often show good results in studies there are always open questions regarding venoms’ potential toxicity on normal non-target cells and tissues making this kind of toxicity one of the highest obstacles when it comes of using venoms in the clinical setting or as medication. There are a large number of papers describing cytotoxic and genotoxic properties of insect venoms as well as biologically active compounds from them.

Garaj-Vrhovac et al. Evaluation of the cytogenetic status of human lymphocytes after exposure to a high concentration of bee venom in vitro. Arh Hig Rada Toksikol. 2009; 60(1): 27-34.

Figure 2 is referenced as opposed to Figure 3. Please explain discrepancy?

Could authors address the issue of bee venom carriers and nanotechnology (nanomedicines) used as a delivery system for bee and wasp venom components? By this approach one may overcome the adverse effects of such venoms when used as a medicine.

Wang et al. Nanomedicines for the treatment of rheumatoid arthritis: State of art and potential therapeutic strategies. Acta Pharm Sin B. 2021; 11(5): 1158-1174.

Yousefpoor et al. Anti-rheumatic activity of topical nanoemulsion containing bee venom in rats. Eur J Pharm Biopharm. 2022; 172: 168-176.

Comments on the Quality of English Language

Minor editing of English language required.

Author Response

Reviewer 2 Report (New Reviewer)

Comments and Suggestions for Authors

Journal: Toxins

Manuscript ID: toxins-3202952
Type of manuscript: Review
Title: Therapeutic Potential of Bee and Wasp Venom in Anti-Arthritic Treatment: A Review

In this manuscript, submitted as a review paper to the Toxins journal, the author(s) analyzed and presented, in an interesting and original approach, the relevance of molecules from bee and wasp venom in the treatment of various types of arthritis. Even though the use of venoms in the treatment of arthritis goes back thousands of years, the subject is of actual scientific interest, proved by the numerous recent publications used for documenting this review: 46% of the used references are published in the last 5 years (2020-2024).

The methodology used is adequate, and the text of the manuscript is cursive and correctly written. The tables and figures used are clear and useful, completing the text.

Therefore, this reviewer recommends the publication of this manuscript into the Toxins journal, but after a minor revision, required to improve its accuracy.

Minor points:

Lines 37-40. The phrase “Over the past decade….” is ended with ref. 10 from 2016. Please replace it with a more recent one.

Lines 43-44. Ref. 13 is not appropriate.

Lines 52-53. Please check the accord.

Lines 52-54. For this phrase, ref. 14 is not appropriate – it is an experimental study on mice. Please find the original article for this statement.

Lines 59-60. Please confirm that Rheum Dis Clin North Am (ref. 18) is a publication belonging to the Chinese Medical Association, or that Ogdie, A., Weiss, P. are members of the Chinese Medical Association. Or rephrase.

Lines 81-83. The authors use the formulation “currently considered to be the most effective treatment strategies” in association with two references from 1990 (ref. 31), and 2006 (ref. 32). Please rephrase or use more recent references. The same comment for ref. 34, from 2020 (lines 86-87).

Lines 105-106. Please rephrase: “Existing treatment options can relieve the indications and manifestations of the condition”.

Line 119. For consistency in enumerating the substances, please replace “enzymes” with “proteins with enzymatic role”.

Line 120. The statement “secreted from the venom sacs of bees” is not correct. Bees have several glands whose cells are involved in synthesis and secretion of various venom molecules that finally accumulate in the venom sac until use. Please rephrase and use a more adequate reference than ref. 54.

Line 136. The authors should include phospholipase A2, due to its high concentration in the BV. And please use a citation for the BV composition (Habermann, who first analyzed and reported the BV composition was not cited in this review).

Line 144. Please use italics for Vespa affinis.

Lines 151-153. Please use references here.

Line 163. Please add “and original articles”.

Lines 195-196. Since the statement “…bee venom at a dose of 20.0 g/kg”. Bee venom treatment 195 did not cause significant changes in hepatotoxicity or liver function indices…” is unbelievable (subcutaneous doses much lower than 100 mg/kg are lethal in rats), the paper “TekeoÄŸlu, İ., AkdoÄŸan, M., Çelik, İ. Investigation of Anti-Inflammatory Effects of Bee Venom in Experimentally Induced Adjuvant Arthritis. Reumatologia 2020, 58, 265-271” was checked. The highest dose of venom used (the third one) in this study was 20 μg/kg (once a week three times) – resulting in a huge difference. Please correct the dose in line 195.

In Table 1, please check the “anti-cancer effects” of apamin.

Line 214. Please replace [69] with [76].

Line 214. Please replace “According to a study in 2018” with “According to the same study from 2018”.

Line 247. Please check whether ref. 83 is appropriate here.

Lines 259-261. Please reorganize the text: “The venom of wasps possesses anti-inflammatory, analgesic, and immunoregulatory properties [14]. Wasp venom possesses anti-inflammatory, analgesic and immunomodulatory proper-ties.”

Line 271. Please replace [65] with [69].

In Table 4, please use italics for Vespa mandarinia.

In Table 4, and in line 300. Please check the statement about mastoparan-M: “accounting for/constituting approximately 70-80% of crude venom”.

Line 288. Please remove “new” or rephrase.

Line 329. Please replace “new” with “scientifically documented”.

Line 557. Please write correctly “Mrna”.

Author Response

Reviewer 3 Report (New Reviewer)

Comments and Suggestions for Authors

In this review, the authors aim to analyze all the literature concerning the application of bee or wasp venom, or their components, for the treatment of arthritis, to synthesize the most relevant articles and report the mechanisms of action that have been described.

Many articles and reviews in the literature concern the therapeutic action of bee venom, less articles have been dedicated to wasp venom; hence, this work could make a significant contribution to this research topic.

However, several errors in the citation and referencing suggest that this text may have been prepared hastily, which could undermine the credibility of the work.

Furthermore, the criteria, that the authors used to decide which articles to consider most relevant in the field, are not clear.

Why do the authors talk about some articles and mechanisms of action and not others? Have the reported mechanisms of action been validated by multiple authors? Can they be considered in a certain way confirmed, or are they all only to be considered preliminary data?

More detailed observations:

Section 3.1. :

the authors summarize articles concerning rats and cells in culture, yet there are many articles in the literature that talk about the effect on humans, why don't the authors report the most relevant of these?

Section 3.2.:

 there are several incorrect citations: - Woon-Hae Kim et al. does not correspond to the article [69]; - Article 76 does not mention IKK; Choe and Kim (2017) does not match the number [79]

The acronym FLS is not specified what it refers to and why these cells are involved in the disease.

Supplementary Table 1 presents text literally copied from UNIPROT: it is sufficient to cite the database, and the method/string used for the search.

The section has many repetitions and does not give a clear view of how reliable the various hypotheses on the mechanisms of action of mellitin are considered.

Section 4.1. Anti-arthritic effects of wasp venom

Article [89] does not talk about ferroptosis but serotonin, and articles [90,91] do not talk about poisons, so these quotes are also incorrect.

Supplementary Table 2 presents text literally copied from UNIPROT: See comments on supplementary Table 1.

The conclusions of the article repeat in summary lines what is reported in the previous paragraphs and do not go so far as to say anything innovative and/or specific.

I strongly recommend a thorough review of all citations and references to ensure accuracy and adherence to journal guidelines. In addition, I suggest establishing a criterion for analysing the vast literature on the subject, and describing the method used in the choice of the articles summarized. The conclusions should include specific suggestions for researchers working in this field, based on literature reviews and not generic literature.

Author Response

This manuscript is a resubmission of an earlier submission. The following is a list of the peer review reports and author responses from that submission.

Round 1

Reviewer 1 Report

Comments and Suggestions for Authors

Respected Authors,

The article's topic is interesting, however, in this form, the article is not acceptable. It doesn’t fill the requirements of a systematic review. There is a lack of a clearly defined research question and protocol (research plan), evidence of a rigorous search process, inclusion and exclusion criteria as well as critical appraisal and evaluation of all included studies.

The introduction focuses on only types of arthritis and does not contain a hypothesis of a study, research question. The systematic review should be prepared according to PRISMA protocol. This part was prepare uncarefully.

Methodology requires improvement.

In part 4 in many places lack of references lines 181, 188,250.

All figures require improvement and references as well as explanation of abbreviations. All tables requires correction and modification.

Last part perspective should be modified as conclusions and should be rewritten.

In all text Authors should pay attention on lowercase and uppercase, italics as well as punctuation marks.

Comments on the Quality of English Language

English language and overall text of manuscript should be modified

Reviewer 2 Report

Comments and Suggestions for Authors

In this review article the author(s) revised several publications related to the treatment of arthritis with bee and wasp venoms and their active components, such as melittin, apamin and several others. This subject of study has great interest in oriental medicine where these principles has shown positive results. In fact, several publications confirm these findings. Thus, this interesting subject of study deserves a deep revision to summarize physicochemical characteristics, mechanisms of action and possible medical importance of the components of these venoms.  Unfortunately, the present review didn’t attend these requirements. It is usually an assemblage of noncritic short sentences gathered information from the literature resulting in phrases disconnected from each other. Here I list some examples:

Line 109: “Several animal medicines have been found to be effective in treating arthritis, including centipedes, scorpions, bees, and wasps [48-50]”.                                               

Lines 128-129: “The main points of this review the main components of bee venom and wasp venom and their mechanism of action in the treatment of arthritis have been listed in charts and text.”    

 Lines 149 – 153: “…here exists a wide variety of bee venoms including melittin (the major active ingredient), apamin, ado-lapin, mast cell-degranulating peptides which are among some of the most active ingredient found within bee venom [59]. Their safety remains an important concern, and the mechanisms of their action and pharmaceutical characterization remain unknown [60]”.                                                                                                     

Line 155: “… represent a diverse array of species, with several being employed in current research endeavors”.

Lines 171-172: “Our future direction will be to conduct in-depth research on the treatment of RA by wasp 171 venom, which will contribute to the research and development of RA drugs”.   

Lines 181- 184: In the study conducted by İbrahim Tekeoğlu et al. (2020), significant improvements in arthritis-related symptoms, as well as inflammatory  markers including IL-1β, TNF-α, and IL-6, total oxidant level (TOL) and oxidative stress  index (OSI) were observed in the group administered with bee venom at a dosage of 20.0  g/kg”.                                       

Lines 234-235: “Inhibition of cyclooxygenase-2 and phospholipase A was discovered to be the mechanism through which APM exerted its anti-inflammatory properties[80]”.                      

Line 243: “Furthermore, the results indicate thast the utilization …”                                  

Lines 255 – 258: “…we created a collagen-induced arthritis (CIA) model and administered wasp venom as treatment during our research. Research has demonstrated that extracts from wasp venom exhibit superior anti-arthritic effects in CIA rats, which can be attributed to their immunomodulatory and anti-inflammatory properties”.   

In fact, as seen on lines 302-306, the author(s) try to use the present manuscript to give credits to their own publications as shown: “In summary, the various compounds present in bee venom and wasp venom can specifically affect arthritis by activating separate signaling pathways, demonstrating impressive effectiveness in experimental animals like rats with adjuvant arthritis and rats with collagen-induced arthritis. Furthermore, our group's recent research has emphasized the innovative discoveries regarding the utilization of wasp venom for managing rheumatoid arthritis.

Furthermore, Tables and Figures do not include appropriate references to support the indicated assumption of targets for the venom principles (Figures) or their mechanisms of action (Tables). Table 1 information on Phospholipase A2 number of amino acid residues and Molecular Mass, must be revised.

It is my conclusion that the manuscript requires a deep revision of English language and conceptual scientific stile of presentation.

Comments on the Quality of English Language

Poor English making sentences very difficult to understanding its meaning.    

Reviewer 3 Report

Comments and Suggestions for Authors

Comments to Authors

v  Title

·         The word (advancements) in the title, is not reflecting the actual current applications of bee venom and wasp venom in in vitro and in vitro and clinical trials, please pay attention to this point!

v  Abstract

  • The authors should add graphical abstract

v  Introduction

  • Please provide some more data about bee venom, its biological activity and therapeutic uses.
  • Line 67: authors mentioned drug used for PsA, try to add them in more detail including their activity and side effects
  • Authors would mention the bio-active compounds of bee venom and bee wasp
  •  Line 156: all the names of wasps must be in italic
  • Mcd, abberivation for?
  • Please kindly, refer to tables and figures in your text by their numbers, not their placement in the text

Round 2

Reviewer 1 Report

Comments and Suggestions for Authors

I would like to underline that Authors made corrections, however they should prepare sytematic review. The article does not fulfill the requirements for a good review paper. The references should not be in italics. New title does not indicate that Authors prepared review work.

Reviewer 2 Report

Comments and Suggestions for Authors

In the revised version the authors attend all my requirements. 

The manuscript can now be accepted for publication.
